# New Insights on the Nutrition Status and Antioxidant Capacity in Multiple Sclerosis Patients

**DOI:** 10.3390/nu11020427

**Published:** 2019-02-18

**Authors:** Ayelet Armon-Omer, Chen Waldman, Naaem Simaan, Hadar Neuman, Snait Tamir, Radi Shahien

**Affiliations:** 1Research Laboratory, Ziv Medical Center, Zefat 1311001, Israel; chen.w@ziv.health.gov.il (C.W.); hadar.n@ziv.health.gov.il (H.N.); 2Department of Neurology, Ziv Medical Center, Zefat 1311001, Israel; naaem.s@ziv.health.gov.il (N.S.); shahien.r@ziv.health.gov.il (R.S.); 3Faculty of Medicine, Bar Ilan University, Zefat 1311502, Israel; 4Department of Medical Laboratory Sciences, Zefat Academic College, Zefat 1320611, Israel; 5Laboratory of Human Health and Nutrition Sciences, MIGAL-Galilee Research Institute, Kiryat Shmona 11016, Israel; snait@telhai.ac.il; 6Nutritional Sciences, Tel-Hai College, Upper Galilee 12208, Israel

**Keywords:** multiple sclerosis, nutrition, nutrients, fatty acids, antioxidant capacity

## Abstract

Background: Multiple sclerosis (MS) is a multifactorial disease with unknown etiology. It is assumed to result from interplay between genetic and environmental factors, including nutrition. We hypothesized that there are differences in nutritional parameters between MS patients and healthy controls. Methods: We examined 63 MS patients and 83 healthy controls. Nutritional status was determined by a dietary questionnaire, blood tests, quantification of cell membrane fatty acids, and serum antioxidant capacity. Results: We found that MS patients consumed a more limited diet compared with the healthy group, indicated by a lower average of 31 nutrients and by consumption levels of zinc and thiamine below the recommended daily intake. Both consumption and measured iron values were significantly lower in MS patients, with the lowest measures in the severe MS group. Long saturated fatty acids (>C16) were significantly lower in MS patients, while palmitic and palmitoleic acids were both higher. Serum total antioxidant capacity was significantly lower in the MS group compared with healthy controls, with the lowest measures in patients with severe MS. Conclusions: This study points to a possible correlation between nutritional status and MS. Understanding the clinical meaning of these findings will potentially allow for the development of future personalized dietary interventions as part of MS treatment.

## 1. Introduction

Multiple sclerosis (MS) is a chronic multifactorial disease of the central nervous system (CNS) and the most common cause of neurological disability in young adults. The clinical presentation of MS is heterogeneous and characterized by a broad spectrum of sensory, motor, cognitive, and neuropsychiatric symptoms. It is caused by damage to the fatty myelin sheath, the protective covering that surrounds nerve cells, leading to malfunction of nerve impulses. The etiology of MS is unknown; it is thought to involve a complex genetic attribute (including the HLA-DR15 haplotype) and environmental factors including Epstein–Barr virus infection [1], low vitamin D3 status [2], and nutrition [3]. 

Nutrition is one of the environmental factors that may have an effect long before MS becomes clinically evident, and for this reason the causal pathways of nutrition are difficult to determine. Generally, countries with high abundance of MS are characterized by high-fat/high-carbohydrate and hypercaloric “Western” diets, typically with high saturated fats and processed foods. Several studies have linked diet and MS [4,5], but there is no clear evidence regarding the efficacy of nutrition as a complementary MS treatment. A recent study found most people with MS did not meet nutritional guidelines for fruit, vegetables, and whole grain consumption [6]. Already in the 1950s, Swank et al. observed that people who consumed less animal fat had lower rates of MS [7]. Studies investigating an MS animal model revealed harmful as well as beneficial effects of saturated fats, depending on their aliphatic chain length [8]. 

The natural antioxidants in diet play an important role in human health. Antioxidants, which are divided into enzymatic and non-enzymatic, are elements that protect the body against free radicals. The levels of melatonin and other antioxidant vitamins were previously shown to be significantly lower in MS patients compared to controls [9]. Recent work found loss of a major brain antioxidant, glutathione, in progressive MS [10]. However, the timing, the degree, and the mechanisms by which antioxidants contribute to MS are still unclear and need to be elucidated. 

As diet can exert direct effects on various cellular elements, influence the gut microbiome, and result in changes in metabolism and immune function, we chose to further investigate dietary parameters in MS patients in this study. 

We hypothesized that MS patients may display nutritional differences compared to healthy controls, since some nutrients such as fatty acids may be important for myelin building, and others may affect neuroinflammation. We assumed that we may see changes in 1. the consumption of essential nutrients, such as vitamins, metals, and minerals; 2. the expression of various nutrients in the blood; 3. the composition of fatty acids; and 4. antioxidant capacity.

To this end, the purpose of the current study was to analyze the detailed intake of nutrients as well as examine the blood nutritional absorption and antioxidant capacity and discuss whether these parameters could potentially contribute to the MS state. 

## 2. Materials and Methods 

Ethics: All subjects gave their informed consent for inclusion before they participated in the study. The study was conducted in accordance with the Declaration of Helsinki, and the protocol was approved by the Ethics Committee of Ziv, NCT 01918501. 

Participants: 146 volunteers were recruited, including 63 MS patients (treated and untreated) diagnosed by a neurologist based on clinical, laboratory, and MRI findings; 83 healthy controls with no known neurological disease as self-reported. MS patients were evaluated according to the Expanded Disability Status Scale (EDSS) and divided into mild (EDSS = 0–3) and severe (EDSS ≥ 3.5) groups. MS patients were recruited from the MS clinic at the Ziv Medical Center, Israel. Our controls included participants from the hospital staff and healthy companions who accompanied the patients. Inclusion criteria: age 18–75; exclusion criteria: other pathologies or immune system disorders. 

Questionnaires: A validated 24-hour dietary recall interview including quantity and types of food was guided by trained dieticians. The validation of the 24-hour recall method has been reported extensively [11]. This method has been used in cohorts around the world including in the Israeli population and was chosen in the Israeli national health and nutrition survey (MABAT) [12]. In the 24-hour recall, the participants are asked to report all food and drinks consumed in the past 24 h. The trained interviewers in our study used the specially developed Israel Food and Food Quantities Guide, which includes photographs that demonstrate different portion quantities and improve the accuracy of recalls. Consumption quantities were compared to the Dietary Reference Intakes (DRIs). Evaluation of 66 dietary parameters was performed using Tzameret software (developed by the Food and Nutrition Administration, Israel Ministry of Health). Additional information was collected by questionnaires and medical records, including age, ethnicity, height, weight, and disease-related variables.

Blood samples: For serum isolation, blood samples were taken by venipuncture into anticoagulant-free tubes at recruitment and centrifuged at 1000× *g* for 10 min. For red blood cell (RBC) separation, blood was taken by EDTA-tubes and samples were isolated immediately by centrifugation at 2500× *g* for 10 min. All samples were stored at −80 °C for subsequent analyses. Researchers were blinded for the status of samples when performing analysis. 

Fatty acid composition was analyzed at the HS-Omega-3 index lab (Omega Metrix Lab, Prof. C. von Schacky, Germany) in thawed and separated RBCs by the omega-3 index methodology. Fatty acid methyl esters were generated from RBCs by acid transesterification and analyzed by gas chromatography (GC2010, Shimadzu Duisburg, Germany) using hydrogen as carrier gas [13]. Twenty-six fatty acids were quantified and validated by a standard mixture of fatty acids. Results were calculated as a percentage of total identified fatty acids after response factor correction.

Total antioxidant capacity (TAC) was evaluated by quantitative colorimetric measurements in serum using the TAC Assay Kit (Abcam). By this method, Cu^2+^ is reduced by an antioxidant to Cu^+^. The resulting Cu^+^ specifically forms a colored complex with a dye reagent. Measure output utilizes a microplate reader (Infinite M200 pro TECAN) at OD570 nm. The color intensity is proportional to TAC in the serum sample. Trolox, a water-soluble vitamin E analog, was used to standardize antioxidants, and all other antioxidants were measured in Trolox equivalents. TAC values were expressed within the linear range of 1.5–1000 mM Trolox equivalents, so that higher values reflect improved antioxidant capacity. Each sample was assayed in duplicates. The mean absorbance value of the blank was subtracted from all standard and sample readings for calculating absorbance. The corrected absorbance values were plotted for each standard as a function of the final concentration of Trolox (0–20 nmol). Trend line equations were calculated based on the standard curve data.

Statistics Analysis: For categorical variables, summary tables are provided giving sample size and absolute frequencies. For continuous variables, summary tables are provided giving arithmetic mean (M) and standard deviation (SD). The independent-sample t-test was performed for comparison of age and BMI between MS and control groups (Table 1), after assessing normality with Kolomorov–Smirnov and Shapiro–Wilk tests. The Mann–Whitney non-parametric test was applied for testing additional differences between control and MS groups (Tables 1–4). Kruskal–Wallis non-parametric tests were used to measure the differences between the control and the EDSS groups and between different treatment groups, due to small sample sizes. A P-value of 5% or less was considered statistically significant. All data were analyzed using SPSS version 24 (SPSS Inc., Chicago, IL, USA). 

## 3. Results

We recruited 146 participants, including 63 MS patients with a stable clinical disease course, and 83 healthy volunteers as a control group. The MS group included 40 mild MS (Mean EDSS = 1.8) and 23 severe MS patients (mean EDSS = 5.2). Mean age at diagnosis among the MS group was 33.9 (±11.2) years, and mean disease duration was 10.4 (±10.4) years. Of the 63 MS patients, there were 15 patients untreated at enrolment time, 24 patients treated with interferon beta (15 Rebif®, 8 Avonex®, and 1 Betaferon®), 10 treated with fingolimod (Gilenya®), and 14 treated with other medications (natalizumab (Tysabri®), with no significant differences in EDSS values. Gender, ethnicity, BMI, and age at enrollment were comparable between groups as seen in Table 1. 

### 3.1. MS Patients Display Significant Lower Intake of Many Nutritional Components

Participants completed a 24-hour dietary recall with the assistance of a dietician and quantity estimation of 66 consumed nutritional components was evaluated. Significant differences were found between both MS groups and healthy controls in their overall daily caloric consumption (Table 2). Moreover, despite the similar BMI values in the MS and healthy groups (Table 1), we found significantly lower dietary intake of 31 components within the MS patient group compared with controls including a 15.1% decrease in calories, a 25.7% decrease in protein, a 19.5% decrease in total dietary fibers, and a 28.4% decrease in cholesterol (Table 2). Included in the less-consumed compounds found in the MS group were 11 amino acids (Table 2), several micronutrients such as calcium (lower by 23.7%), iron (27.35%), magnesium (32.3%), phosphorus (25.3%), potassium (25.4%), sodium (26.9%), zinc (29.5%), and copper (31.8%), as well as B vitamins including thiamine (26.8%), riboflavin (27.6%), niacin (19.7%), vitamin B6 (27%), and folate (24.1%). Zinc and thiamin values were significantly lower than the DRI recommendations. Consumption of only one fatty acid (Caprylic C8_0) was significantly different between MS patients and the control group (Table 2). When comparing data obtained from mild and severe MS patients, we observed that the severe MS group showed a significantly higher consumption of linolenic acid (ALA, C18_3n3), the precursor of the omega-3 family (Table 2). However, we did not observe significant differences in the EPA or DHA, which comprise the omega-3 index, although there was a trend towards lower omega-3 values in MS patients as compared to controls. As was the case in [14], we did not find an association between antioxidant intake of relevant vitamins (A, C, D, E, and carotene) and MS disease progression. When comparing between MS patient groups according to treatments, no significant differences were found regarding intake of nutritional components (Appendix A).

### 3.2. Routine Blood Tests Show Lower Iron in MS Patients 

Analysis of routine blood tests revealed that the MS group had significantly lower iron values, as compared to healthy controls (Table 3). When comparing data obtained from mild and severe MS patients, severe MS patients showed even lower iron levels, although not significantly. No significant differences were found in blood iron levels when comparing between MS patient groups according to treatments (see Appendix A).

### 3.3. MS Patients Present Unique Fatty Acid Profiles on Cell Membranes

Biochemical analysis of fatty acid profiles from membranes of RBC samples was performed using gas chromatography. To our knowledge, this was the first study to be done on MS samples using this technique. We received significant differences in a number of fatty acids when comparing overall MS patients with healthy controls (Table 4). Long saturated fats (C18_0, C20_0, C22_0, and C24_0) were all significantly lower in MS patients compared with controls, and even more so in patients with severe MS. The only saturated fat that was higher in MS patients compared with controls was C16_0, palmitic acid, the most abundant saturated fatty acid in animals (Table 4). Regarding monounsaturated fats, palmitoleic acid (C16_1n7) was found to be significantly higher in severe MS patients compared with controls. On the other hand, C24_1n9 was significantly lower in MS patients and even lower in patients with severe MS (Table 4). We did not find statistically significant differences in the omega-3 or omega-6 levels between MS patients and healthy controls (Table 4). Additionally, no significant differences were found in fatty acid profiles when comparing between MS patient groups according to treatments (see Appendix A). 

### 3.4. Antioxidant Capacity Is Limited in MS, Especially in the Severe State

Non-enzymatic antioxidants classify into low molecular weight (including uric acid, vitamins C, D, and E, glutathione, coenzyme Q, and β-carotene) and ions such as iron, copper, zinc, and manganese. Since antioxidants work in an antioxidant complex to exert protective effects, no single antioxidant can represent overall antioxidant status. Therefore, we chose to assess serum TAC using enzymatic and non-enzymatic segments of the antioxidant defense system. Lower levels of TAC were found in serum of MS patients compared with healthy controls, with significantly low (and the lowest) values in patients with severe MS (Figure 1). 

Altogether, we found an array of differences in nutritional parameters between MS patients and healthy controls. MS patients consumed lower levels of essential nutrients, exhibited distinct fatty acid profiles, and displayed decreased antioxidant capacity. 

## 4. Discussion

### 4.1. Differences in Consumption Levels of Nutritional Components Are Found between MS Patients and Healthy Controls

We found significant differences in the intake of 31 nutritional components (47% of all measured components), with lower levels found in MS patients compared with healthy controls. The MS group showed lower dietary intake of food energy, total dietary fibers, cholesterol, total protein, and certain micronutrients and vitamins. These dietary reports in MS patients describe the current state of disease and possibly an awareness of their health status, and do not necessarily reflect previous potential intake before diagnosis. Additionally, patients may consume less food due to disability problems, including difficulty in swallowing and/or mood influence on appetite. It is, therefore, possible that regardless of MS there is a link between dietary data and physical disability. Importantly, it is impossible to relate the alterations seen in this study in MS patients to specifically cause or result of the disease state, as they are purely correlational. Moreover, since most of the lower consumed components in MS patients were still within the range of the DRI guidelines, they may not necessarily reflect a dietary problem or pathology. Finally, it is important to recognize the limitations of questionnaire data in terms of accuracy, as some participants may not report their exact intake for various reasons. Moreover, we did not observe lower BMI levels or lower albumin levels in the patient group, suggesting that, despite lower intake in patients compared with controls, patients did not exhibit malnutrition by standard measures [15,16].

### 4.2. MS Patients Display Low Levels of Essential Minerals

We found significant reduced nutritional intake of several essential minerals in the MS group, including zinc, magnesium, iron, and copper. This is of particular interest as metal homeostasis plays a critical role in regulation of the CNS and is essential for normal functioning. Moreover, metal deficiencies have been found in serum of MS patients, specifically iron, magnesium, lithium, and zinc [17,18,19,20], and metal imbalances have been linked to demyelination, perhaps involving oxidative stress [21]. The zinc mean consumption value was 29.5% lower in the MS group compared with controls, below the DRI recommendation. These findings are consistent with lower serum zinc levels in MS patients previously published [20,22]. Of relevance, in the CNS, zinc plays roles in modifying neuronal excitability and synaptic plasticity. Magnesium was also significantly lower in MS patient intake (by 32.3% compared to controls), as was found in a recent study [23]. This is of particular interest since magnesium is known to improve neurological outcome in patients with neural damage [24], although the exact neuroprotection mechanisms remain uncertain. Additionally, a published post-mortem study showed that magnesium concentrations were significantly lower in CNS tissue obtained from MS patients compared to controls, and the greatest decline of magnesium content was observed in the demyelinated plaques of MS patients [25]. Iron blood levels of MS patients were also significantly lower from controls, as was the dietary iron intake (by 27.35%). When comparing data obtained from mild and severe MS patients, the severe MS group showed even lower iron. This result is in agreement with recent work suggesting that MS patients show decreased blood iron concentrations over time [26] as well as decreased iron intake [23]. Iron is considered an important factor in the pathogenesis of MS [27], as it may cause neuronal damage by stimulating oxidative stress. Pediatric MS cases were shown to consume insufficient iron compared to controls [28]. Some studies suggest that iron insufficiency may play a role in MS disease progression as MS patients display clinical improvement upon iron supplementation. However, other studies indicate improved disease outcome in iron-limited MS patients [29]. These contradicting results may be due to differences in nutritional and biochemical factors between subjects, requiring further investigation. Alternatively, it is possible that inadequate iron levels (both low and high) may be harmful in MS since iron excess might increase free radicals, which may elevate oxidative stress, while iron reduction could decrease immune system function and cause an energy deficit due to loss of mitochondria membrane potential [29]. Finally, we found lower dietary copper intake in the MS group (lower by 31.8%). Copper is an essential cofactor for many oxidative enzymes and is necessary for iron absorption and transfer. This fits the experimental animal model of MS in which administration of cuprizone as a copper chelator leads to pathology. The cuprizone toxicity affects oligodendrocytes, probably by causing mitochondrial injury, oxidative stress, and subsequent apoptosis [30]. In contrast, correlation between MS and high free copper levels in serum and cerebrospinal fluid has been previously suggested [31]. The precise roles of copper in MS pathology require further investigation.

### 4.3. MS Patients Display Low Intake Levels of B Vitamins

We found significant reduced nutritional intake of B vitamins in MS patients including riboflavin (27.6%), niacin (19.7%), vitamin B6 (27%), and folate (24.1%), matching other studies [23]. Thiamin mean consumption was lower in the MS group (26.8%) and below recommended guidelines. Several human studies support the protective role of B vitamins in MS incidence and progression [32]. For example, thiamine deficiency causes increased chemokine ligand2 expression in neurons of the MS mice model [33]. No significant differences in B12 values were found between groups, as seen in other studies [34]. This data complied with B12 levels we measured according to the consumption questionnaire. 

### 4.4. The Fatty Acid Profiles on Cell Membranes in MS Patients Are Different from Controls

Since the brain is an organ with high lipid concentrations, its fatty acid composition is of specific interest in understanding mechanisms of neurological diseases, including MS. Fatty acids, as structural components of membranes and inflammatory/anti-inflammatory mediators, have important well-known regulatory effects [35]. Since direct access to CNS lipids is not available, we chose to test the fatty acid profiles from RBC membranes, reflecting the dietary fatty acid intake averaged over the RBC lifespan of up to 120 days. Out of the 26 measured fatty acids, 9 showed significant differences between groups; MS patients presented lower levels of 6 fatty acids and higher levels of 3 fatty acids, as compared to controls. As the actual value differences between group means were small even when significant, we cannot be sure regarding their clinical meaning or relevance. Moreover, even if the observed differences in fatty acid profiles of MS patients are pathological, it is still unclear whether they are just a phenomenon of other pathogenic mechanisms or rather actually contributes to the pathogenesis of MS. Nonetheless, there are some indications in the literature that the differences found may have physiological meaning [5,35]. The only saturated fat that appeared at higher levels in the MS patient group compared with control was palmitic acid (C16_0), the most abundant saturated fatty acid accounting for 20–30% of total fatty acids in the human body. In mammals, palmitic acid endogenous biosynthesis may not be sufficient, so adequate levels may depend on optimal intake of unsaturated fatty acids [36]. This finding fits a recent lipidomic profiling study in which MS patients were found to have increased cerebrospinal fluid levels of ceramide C16:0, sufficient to induce axonal damage [37]. Another work found a significant increase of C16:0 sulfatide in extracellular vesicles isolated from plasma of MS patients compared with healthy controls [38]. Palmitoylation is a reversible post-translational modification of proteins that consists in a covalent attachment of a palmitic acid to specific cysteine via a thioester bond. This is of particular interest as a current review demonstrated that palmitoylation might be critically involved in the regulation of neurotransmitter receptor functions [39]. Levels of monounsaturated palmitoleic acid (16_1n7) were shown to be significantly higher in severe MS patients compared with healthy controls. A similar finding was previously reported for Behcet Disease, in which patients with the severe form of disease also had higher concentrations of palmitoleic acid [40]. Palmitoleic acid is synthesized by the desaturation of palmitic acid via stearoyl-CoA desaturase 1 activity. It acts as a signaling lipid hormone and is produced and secreted mostly by white adipose tissue [41]. It has the capacity to serve as a lipid signal that mediates communications between adipose and other tissues [42]. In animal models, palmitoleic acid reduces the expression of proinflammatory markers that are related to metabolic abnormalities [43]. While recent work revealed changes in another monounsaturated fat, oleic acid, in MS post-mortem brains and in animals [30], we did not observe similar changes in our study. Among the longer monounsaturated fats, we found significantly low values of C24_1n9 in MS patients, with the lowest values observed in patients with severe MS. Interestingly, most of the long fatty acids measured here had lower values in MS patients, whereas fatty acids with 16 carbons (saturated and monounsaturated) were higher in the MS patient group. 

We did not find a difference in the omega-3 intake nor in RBC membrane values between the MS and healthy groups, although we found a tendency toward lower omega-3 values in MS patients, with the lowest levels found in patients with severe MS. Here we found that the MS group had a significantly lower dietary intake of only one saturated fat, caprylic acid (C8_0), as compared to the healthy group. While we did not find any information regarding caprylic acid and MS in the literature, it would be interesting to further investigate whether low levels are involved in MS pathogenesis. 

### 4.5. Decreased Antioxidant Capacity Is Associated with MS Disease Severity

A significantly lower value of TAC was observed in MS patients, as compared to the healthy group, conferring published data [44] indicating that diet antioxidants can protect against oxidative damage and related inflammatory complications [45]. Moreover, significantly lower levels of TAC were observed in severe compared with mild MS patients. The poor antioxidant capability in the severe form of MS may play a role in disease progression. Despite these significant results, we did not find a correlation between dietary metabolites and antioxidant capacity. Antioxidants may have a positive impact on the course of MS. A number of new studies found that dimethyl fumarate, natalizumab, and fingolimod therapies have a positive effect on antioxidant capacity. 

## 5. Conclusions

MS is a chronic disorder with uncertain pathogenesis. To our knowledge, this is the first study to combine nutritional data, antioxidant capacity, and fatty acid profiles in MS patients. We found significant differences in nutritional components (as measured in blood and assessment of consumption) between MS and healthy controls and between mild and severe MS patients. MS patients showed significantly lower levels of long-chain saturated fats and long monounsaturated fats in blood as well as lower food intake including proteins, vitamins, minerals, and caprylic acid. On the other hand, the only saturated fat that was high in our MS group was palmitic acid. Moreover, palmitoleic acid was significantly higher in the severe MS group. Additionally, we found MS patients had decreased levels of serum total antioxidant capacity.

As this study focused on nutritional aspects that can potentially be modified by diet or lifestyle, future research may lead to promising avenues of therapy for MS. Our findings point to antioxidant and nutritional directions to be considered for future research. This research may serve as a model for other neurological diseases since MS can be considered a case of “micro brain damage”. 

## Figures and Tables

**Figure 1 nutrients-11-00427-f001:**
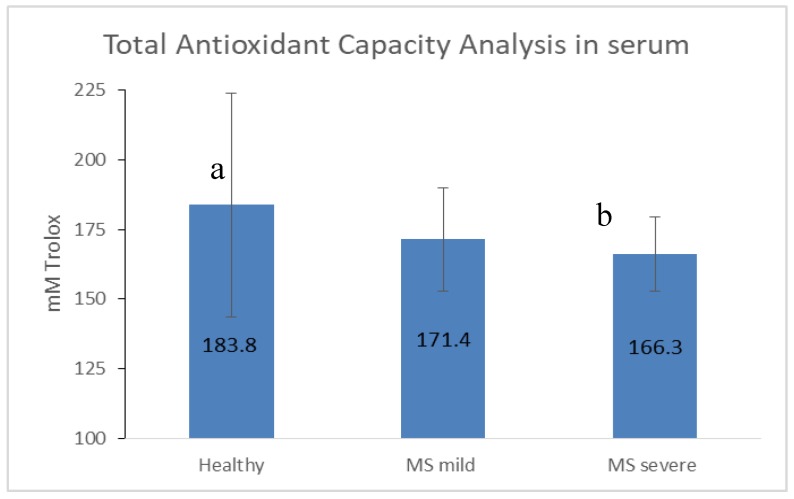
Total antioxidant capacity in serum. MS: multiple sclerosis; mild: EDSS 0-3; severe: EDSS ≥ 3.5. Values are reported as mean ± SD. Different letters indicate significant differences between groups.

**Table 1 nutrients-11-00427-t001:** Demographic variables of the study participants.

Variables	Values	Control (*n* = 83)	MS (*n* = 63)	*p*
Age at enrolment, year (M ± SD)		40.6 ± 11.9	44.7 ± 14.0	0.056
Gender (*n*, %)	Female	49, 59.0	42, 66.7	0.346
Ethnicity (*n*, %)	Jew	54, 65.1	41, 65.1	0.998
BMI, kg/h^2^ (M ± SD)		25.3 ± 4.7	25.0 ± 4.4	0.732

Abbreviations: multiple sclerosis (MS), mean (M), standard deviation (SD), count (*n*), body mass index (BMI).

**Table 2 nutrients-11-00427-t002:** Daily consumption of 66 dietary items in MS patients vs. healthy controls and in severe vs. mild MS patients.

Daily Consumption Data (/day)	Control	MS	*p* *	MS		*p* **
	Mean	Mean		EDSS 0–3	EDSS ≥ 3.5	
**General Components**						
Food energy (Kcal)	1973.91 ^a^	1675.62	**0.013**	1627.56 ^b^	1757.33 ^ab^	0.036
Protein (g)	87.07 ^a^	64.70	**0.000**	65.22 ^b^	63.81 ^b^	**0.001**
Carbohydrates (g)	224.79	196.73	0.108	195.09	199.38	0.273
Moisture (g)	3400.68 ^a^	2279.22	**0.000**	2370.08 ^b^	2124.76 ^b^	**0.001**
Total Dietary Fibers (g)	25.24 ^a^	19.53	**0.008**	18.42 ^b^	21.32 ^ab^	**0.019**
Alcohol (g)	1.13	0.35	0.210	0.57	0.00	0.380
EER (kcal)	2144.9	2009.3	0.341	2074.6	1903.6	0.456
Total Sugars (g)	66.16	60.73	0.455	55.49	69.21	0.327
**Minerals**						
Calcium (mg)	830.17 ^a^	633.07	**0.012**	609.44 ^b^	673.26 ^b^	**0.036**
Iron (mg)	14.70 ^a^	10.69	**0.008**	10.08 ^b^	11.75 ^b^	**0.023**
Magnesium (mg)	519.80 ^a^	351.73	**0.003**	341.18 ^b^	369.68 ^b^	**0.014**
Phosphorus (mg)	1338.61 ^a^	999.57	**0.000**	1012.88 ^b^	976.95 ^b^	**0.001**
Potassium (mg)	3339.78 ^a^	2490.56	**0.000**	2434.61 ^b^	2590.67 ^b^	**0.001**
Sodium (mg)	3274.69 ^a^	2392.66	**0.002**	2321.24 ^b^	2520.47 ^ab^	**0.006**
Zinc (mg)	10.66 ^a^	7.51	**0.000**	7.68 ^b^	7.20 ^b^	**0.000**
Copper (mg)	2.11	1.44	0.051	1.29	1.67	0.112
**Vitamins**						
Vitamin A (µg)	562.08	467.94	0.273	481.67	443.52	0.527
Vitamin E (mg)	7.97	7.37	0.530	7.39	7.32	0.821
Vitamin C (mg)	112.18	101.71	0.517	99.10	105.94	0.778
Vitamin D (µg)	46.29	30.52	0.230	30.77	30.19	0.512
Thiamin (B1) (mg)	1.27^a^	0.93	**0.001**	0.89^b^	0.99 ^ab^	**0.005**
Riboflavin (B2) (mg)	2.10 ^a^	1.52	**0.003**	1.46 ^b^	1.63 ^ab^	**0.009**
Niacin (B3) (mg)	24.36	19.56	**0.045**	18.10	21.92	0.072
Vitamin B6 (mg)	1.96	1.43	**0.000**	1.41 ^b^	1.47 ^b^	**0.001**
Folate B9 (µg)	359.75 ^a^	273.06	**0.003**	259.42 ^b^	296.26 ^ab^	**0.010**
Vitamin B12 (µg)	3.56	2.74	0.079	2.59	2.99	0.182
Carotene (µg)	3175.02	2936.18	0.682	3417.06	2075.67	0.283
**Fats**						
Total fat (g)	78.88	67.39	0.091	62.98	74.90	0.117
Cholesterol (mg)	310.75	222.49	**0.042**	214.92	235.36	0.121
Saturated fat (g)	25.41	21.39	0.114	20.54	22.83	0.240
Trans fatty acids (g)	0.21	0.26	0.737	0.32	0.16	0.756
Butyric C4_0 (g)	0.51	0.43	0.335	0.44	0.42	0.624
Caproic C6_0 (g)	0.29	0.23	0.163	0.25	0.21	0.307
Caprylic C8_0 (g)	0.28	0.20	**0.042**	0.20	0.19	0.123
Capric C10_0 (g)	0.47	0.36	0.098	0.35	0.37	0.251
Lauric C12_0 (g)	0.82	0.52	0.167	0.47	0.60	0.354
Myristic C14_0 (g)	2.05	1.62	0.098	1.61	1.63	0.256
Palmitic C16_0 (g)	12.36	10.87	0.243	10.92	10.77	0.498
Stearic C18_0 (g)	5.56	4.66	0.174	4.46	5.01	0.340
Oleic C18_1n9 (g)	26.31	22.00	0.091	22.30	21.46	0.236
Linolenic ALA C18_3n3 (g)	1.69 ^ab^	1.61	0.680	1.31 ^b^	2.07 ^a^	**0.026**
Arachidonic C20_4n6 (g)	0.12	0.10	0.243	0.09	0.12	0.249
Docosahexanoic DHA C22_6n3 (g)	0.11	0.10	0.830	0.08	0.11	0.901
Palmitoleic C16_1n7 (g)	1.10	0.97	0.349	0.95	0.99	0.635
Parinaric C18_4n3 (Conjugated) (g)	0.01	0.00	0.179	0.01	0.00	0.368
Gadoleic C20_1n11 (g)	0.29	0.38	0.463	0.36	0.39	0.748
Eicosapentaenoic EPA C20_5n3 (g)	0.03	0.03	0.973	0.02	0.04	0.873
Erucic C22_1n9 (g)	0.04	0.03	0.530	0.03	0.04	0.755
Docosapentaenoic DPA C22_5 (g)	0.02	0.02	0.713	0.02	0.02	0.921
Monosaturated (g)	30.79	25.32	0.174	22.64	29.87	0.190
Polysaturated (g)	12.30	9.48	0.069	8.94	10.37	0.159
**Amino Acids**						
Isoleucine (g)	1.80	1.30	**0.029**	1.22	1.42	0.077
Leucine (g)	3.14	2.21	**0.019**	2.10	2.39	0.056
Valine (g)	2.10 ^a^	1.50	**0.017**	1.42 ^b^	1.61 ^ab^	**0.050**
Lysine (g)	2.80	1.94	**0.030**	1.83	2.13	0.082
Threonine (g)	1.59	1.13	**0.026**	1.07	1.24	0.072
Methionine (g)	0.92	0.66	**0.032**	0.64	0.69	0.098
Phenylalanine (g)	1.82 ^a^	1.31	**0.017**	1.24 ^b^	1.42 ^b^	**0.049**
Tryptophan (g)	0.27	0.22	0.348	0.22	0.23	0.635
Histidine (g)	0.98	0.74	0.098	0.66	0.88	0.148
Tyrosine (g)	1.39	0.97	**0.018**	0.93	1.05	0.053
Arginine (g)	2.34	1.65	**0.028**	1.48	1.92	0.055
Cystine (g)	0.56	0.41	**0.041**	0.38	0.46	0.092
Serine (g)	1.86 ^a^	1.32	**0.013**	1.26 ^b^	1.42 ^ab^	**0.041**

Abbreviations: multiple sclerosis (MS), Expanded Disability Status Scale (EDSS), mean (M), mild MS (EDSS 0–3), severe MS (EDSS ≥ 3.5), gram (g), estimated energy requirement (EER), alpha linolenic (ALA), docosahexanoic (DHA), eicosapentaenoic (EPA), docosapentaenoic (DPA). * control vs. MS. ** control vs. two groups of disease severity (evaluated by EDSS). Values of daily consumption are reported as M ± SD. Bold Values show a significant difference. Values with different letters superscript in the same row are significantly different between groups.

**Table 3 nutrients-11-00427-t003:** Routine blood test values comparing MS patients vs. healthy controls and severe vs. mild MS patients.

Variables	Control	MS	*p* *	EDSS 0–3	EDSS ≥ 3.5	*p* **
Iron (µg/dL)	78.7 ± 33.0	62.7 ± 35.3	**0.043**	69.9 ± 35.9	50.2 ± 31.8	0.054
Ferritin (ng/mL)	111.2 ± 93.8	78.9 ± 70.4	0.105	65.5 ± 62.0	101.3 ± 80.4	0.155
Vitamin B12 (pg/mL)	417.2 ± 144.9	447.9 ± 170.4	0.340	451.6 ± 164.6	440.8 ± 186.0	0.859
Magnesium (mg %)	2.0 ± 0.2	2.1 ± 0.1	0.377	2.2 ± 0.1	2.0 ± 0.1	0.066
Folic acid (ng/mL)	9.3 ± 4.0	10.0 ± 5.2	0.473	9.5 ± 4.9	10.9 ± 5.8	0.537
Triglycerides (mg/dL)	110.7 ± 64.3	109.0 ± 65.2	0.895	104.8 ± 61.2	117.3 ± 70.7	0.757
Total Cholesterol (mg/dL)	186.5 ± 46.6	178.2 ± 32.1	0.265	172.9 ± 26.5	186.6 ± 38.8	0.493
HDL (mg/dL)	50.5 ± 11.5	48.9 ± 13.1	0.515	46.7 ± 10.2	53.1 ± 17.0	0.197
LDL (mg/dL)	110.6 ± 37.7	109.5 ± 27.3	0.869	104.7 ± 25.6	118.8 ± 28.7	0.277
Albumin (g %)	4.1 ± 0.5	4.0 ± 0.5	0.303	4.1 ± 0.4	3.9 ± 0.5	0.106

Abbreviations: multiple sclerosis (MS), Expanded Disability Status Scale (EDSS), low-density lipoprotein (LDL), high-density lipoprotein (HDL). * control vs. MS; ** control vs. two groups of EDSS. Values of routine blood test are reported as mean ± SD. Bold Values show a significant difference.

**Table 4 nutrients-11-00427-t004:** Fatty acid profiles from membranes of red blood cells (RBCs) comparing MS patients vs. healthy controls and mild vs. severe MS patients.

Variables			Control	MS	*p* *	EDSS 0–3	EDSS ≥ 3.5	*p* **
**Saturated**		**Myristic C14_0 **	0.25 ± 0.10	0.27 ± 0.15	0.399	0.25 ± 0.14	0.30 ± 0.16	0.283
		**Palmitic acid C16_0**	21.2 ±1.0 ^b^	21.7 ± 1.1	**0.024**	21.5 ± 1.0 ^ab^	21.9 ± 1.2 ^a^	**0.023**
		**Stearic C18_0**	17.3 ± 1.4 ^a^	16.7 ± 1.9	**0.036**	16.9 ± 1.9 ^ab^	16.3 ± 2.1 ^b^	**0.048**
		**Eicosanoic C20_0**	0.20 ±0.06 ^a^	0.16 ± 0.05	**0.000**	0.17 ± 0.05 ^b^	0.15 ± 0.05 ^b^	**0.000**
		**Docosanoic C22_0**	0.41 ± 0.13 ^a^	0.31 ± 0.14	**0.000**	0.32 ± 0.13 ^b^	0.28 ± 0.15 ^b^	**0.000**
		**Lignoceric C24_0**	1.11 ± 0.40 ^a^	0.93 ± 0.36	**0.010**	0.98 ± 0.35 ^ab^	0.85 ± 0.37 ^b^	**0.015**
**Mono** **saturated**		**Palmitoleic C16_1n7**	0.21 ± 0.16 ^b^	0.27 ± 0.22	0.066	0.23 ± 0.19 ^ab^	0.34 ± 0.25 ^a^	**0.019**
		**Oleic C18_1n9**	14.6 ± 1.4	14.9 ± 1.5	0.200	14.8 ± 1.4	15.1 ± 1.6	0.291
		**Eicosenoic C20_1n9**	0.27 ± 0.06	0.25 ± 0.06	0.199	0.25 ± 0.07	0.25 ± 0.04	0.404
		**Nervonic C24_1n9**	1.12 ± 0.33 ^a^	0.97 ± 0.40	**0.028**	1.02 ± 0.41 ^ab^	0.89 ± 0.39 ^b^	**0.033**
**Polyunsaturated**	**Omega6**	**Linoleic C18_2n6**	12.8 ± 3.1	13.1 ± 3.1	0.646	12.8 ± 3.1	13.4 ± 3.2	0.718
		**gamma Linolenic C18_3n6**	0.10 ± 0.06	0.10 ± 0.06	0.868	0.09 ± 0.06	0.11 ± 0.06	0.575
		**Eicosadienoic C20_2n6**	0.31 ± 0.06	0.30 ± 0.06	0.347	0.30 ± 0.06	0.30 ± 0.05	0.644
		**DGLA C20_3n6**	1.75 ± 0.49	1.69 ± 0.30	0.406	1.67 ± 0.30	1.70 ± 0.30	0.688
		**Arachidonic C20_4n6**	16.6 ± 1.9	16.5 ± 2.1	0.856	16.5 ± 2.0	16.6 ± 2.2	0.963
		**Docosatetraenoic C22_4n6**	3.52 ± 1.03	3.65 ± 1.13	0.504	3.7 ± 1.1	3.6 ± 1.2	0.765
		**Docosapentaenoic C22_5n6**	0.97 ± 0.34	1.02 ± 0.33	0.379	1.01 ± 0.34	1.04 ± 0.32	0.649
	**Omega3**	**ALA C18_3n3 **	0.15 ± 0.09	0.16 ± 0.11	0.292	0.15 ± 0.12	0.18 ± 0.11	0.296
		**EPA C20_5n3 **	0.42 ± 0.23	0.50 ± 0.38	0.131	0.56 ± 0.44	0.42 ± 0.25	0.071
		**Docosahexaenic C22_5n3**	1.86 ± 0.44	1.96 ± 0.50	0.255	2.05 ± 0.53	1.80 ± 0.41	0.067
		**DHA C22_6n3 **	4.31 ± 1.16	4.13 ± 1.43	0.436	4.3 ± 1.50	3.92 ± 1.37	0.457
		**Omega3 Index**	4.76 ± 1.32	4.40 ± 1.59	0.214	4.47 ± 1.65	4.29 ± 1.54	0.419
**Trans**		**C16_1n7t trans**	0.07 ± 0.02	0.06 ± 0.03	0.068	0.06 ± 0.03	0.06 ± 0.03	0.186
		**C18_1t trans**	0.35 ± 0.12 ^a^	0.29 ± 0.14	**0.007**	0.28 ± 0.13 ^b^	0.29 ± 0.15 ^b^	**0.026**
		**C18_2n6tt trans**	0.03 ± 0.07	0.03 ± 0.04	0.442	0.04 ± 0.05	0.03 ± 0.04	0.714
		**C18_2n6ct trans**	0.02 ± 0.02 ^b^	0.03 ± 0.02	**0.006**	0.02 ± 0.02 ^b^	0.03 ± 0.03 ^a^	**0.011**
		**C18_2n6tc trans**	0.09 ± 0.03	0.09 ± 0.03	0.698	0.09 ± 0.03	0.10 ± 0.04	0.223

Analysis of fatty acids was performed via gas chromatography; values are reported as mean ± SD. Abbreviations: alpha linolenic (ALA); dihomo-gamma-linolenic acid (DGLA); omega-3 index was calculated as the sum of eicosapentaenoic acid (EPA) and docosahexaenoic acid (DHA); multiple sclerosis (MS); Expanded Disability Status Scale (EDSS). * Control vs. MS; ** control vs. two groups of EDSS. Values with different letters superscript in the same row are significantly different between groups. Bold values show a significant difference.

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
