# Peer review of "New Insights on the Nutrition Status and Antioxidant Capacity in Multiple Sclerosis Patients"

_nutrients, 2019, doi:10.3390/nu11020427_

Reviewer 1 Report

The paper by Armon-Omer et al, compares  the nutritional differences between 63 multiple sclerosis individuals and 83 "age and gender matched" healthy controls.  I believe the article warrants publication but have some concerns to make about the paper.

Major Concerns

Materials and Methods 

Table 1 shows MS patients with a average age of 40.6 +/- 11.9 years were compared to healthy controls with a mean age of 44.7+/- 14.0 years resulting in a p value of 0.056.  This table also shows 49 MS females were compared to 42 control females with a p value of 0.346. There would therefore be 34 MS males compared to 21 healthy control males with no statistical comparisons. I would not consider this an age and gender matched study.  Please state reasons for considering this an age and gender matched study.

Table 1 does not list the number of MS participants using treatment.  It might be useful to include the number of MS individuals undergoing MS treatment (and the type of treatment being used). This would be helpful to the reader to see if there are any interactions with medication treatment and absorption of nutrients being tested. If this is unknown, then state it is a limitation to the study.

What was the criteria for selecting controls?  How were they labeled as healthy? 

Minor Concerns

Reference needed in Introduction after sentence "Studies investigating an MS animal model revealed harmful as well as beneficial effects of saturated fats depending on their aliphatic chain length".

References needed in Discussion: The fatty acid profiles on cell membranes in MS patients are different from controls, after sentence "Nonetheless, there are some indications in the literature that the differences found may have physiological meaning"

Author Response

Dear Reviewer,

We would like to thank you for your meaningful comments and suggestions. We have addressed the issues raised and have revised and improved the manuscript accordingly. In general, we have expanded the materials and methods section, have added data on patient treatments, have added more references to strengthen our discussion, and have rephrased some sentences to improve the fluency and language.  Please see specific answers to your comments:

Table 1 shows MS patients with an average age of 40.6 +/- 11.9 years were compared to healthy controls with a mean age of 44.7+/- 14.0 years resulting in a p value of 0.056.  This table also shows 49 MS females were compared to 42 control females with a p value of 0.346. There would therefore be 34 MS males compared to 21 healthy control males with no statistical comparisons. I would not consider this an age and gender matched study.  Please state reasons for considering this an age and gender matched study.

The age range in both groups was 19-71 years of age.  P-value of the gender distribution was performed as commonly done by Chi-square correlation.

We have changed our phrasing to now not include the term “age-matched”, as of not to mislead.

Table 1 does not list the number of MS participants using treatment.  It might be useful to include the number of MS individuals undergoing MS treatment (and the type of treatment being used). This would be helpful to the reader to see if there are any interactions with medication treatment and absorption of nutrients being tested. If this is unknown, then state it is a limitation to the study. We have added data regarding the MS treatments of study participants in Results section 3.1. We now present in the text and in Supplementary Table 1 that analysis according to treatment did not result in significant differences between groups.

What was the criteria for selecting controls?  How were they labeled as healthy? We have more clearly stated that the healthy control group included individuals between 18-75 years old with no known neurological disease as self-reported by participants. As in any study- we cannot be certain that they do not have a disease not yet diagnosed. 

Reference needed in Introduction after sentence "Studies investigating an MS animal model revealed harmful as well as beneficial effects of saturated fats depending on their aliphatic chain length". An appropriate reference was added.

References needed in Discussion: The fatty acid profiles on cell membranes in MS patients are different from controls, after sentence "Nonetheless, there are some indications in the literature that the differences found may have physiological meaning" We have added appropriate references.

 Reviewer 2 Report

This study investigates the nutrition status and antioxidant 2 capacity in multiple sclerosis patients. Several differences were observed between control and MS patients, and the authors conclude that “This study points to a 26 possible link between nutritional status and MS.”

The described data are purely descriptive, and the conclusion that there is a causal relation would not be justified by the presented data. This limitation should clearly by pointed out, even in the abstract.

Other comments are:

1. Please carefully check the entire manuscript for typo such as line 38 “)”

2. Please provide a citation for “Studies investigating an MS animal model revealed harmful as well as beneficial effects of saturated fats depending on their aliphatic chain length.”

3. Please correct the statement “Serum and red blood cells (RBCs) were immediately centrifuged” as this sentence appears to be incomplete

4. The material and method section is short, and most procedures are not explained in detail. Please provide a more detailed description of the performed experiments to increase reproducibility of the results.

5. Did the authors check whether their data show a normal distribution? If so, which test was performed.

6. The study design is not without risk. The authors compare nutritional intake between MS patients and healthy controls. However, the clinical disability per se might be an important variable for the study outcome. It would have been elegant to include a group with comparable level of clinical disability suffering from another neurological disorder than MS. However, I am aware that this is a complicated study design. At least the authors should discuss this point.

7. To what extent is a 24h dietary recall “representative/meaningful”

8. Please check lines 157-160.

9. Please provide information regarding the patient’s treatment. Did the treatment affect the outcome of the study?

10. The authors found lower iron levels in MS patients versus control. However, as stated correctly, Iron is considered an important  factor in the pathogenesis of MS, as it may cause neuronal damage by stimulating oxidative stress. This discrepancy should be discussed more in detail.

11. With respect to copper levels, a brief discussion of the cuprizone model would be beneficial.

12. Please include a recent study (J Neurochem. 2018 Jun;145(6):504-515) addressing the lipid profile in a preclinical MS model.

Author Response

Dear Reviewer,

We would like to thank you for your meaningful comments and suggestions. We have addressed the issues raised and have revised and improved the manuscript accordingly. In general, we have expanded the materials and methods section, have added data on patient treatments, have added more references to strengthen our discussion, and have rephrased some sentences to improve the fluency and language.  Please see specific answers to your comments:

1. The described data are purely descriptive, and the conclusion that there is a causal relation would not be justified by the presented data. This limitation should clearly by pointed out, even in the abstract. Indeed, from this study it is not possible to draw a conclusion of cause and effect, only about association. We did not intend to suggest any other interpretation of results. This has been clarified in the Abstract and Discussion.

2. Please carefully check the entire manuscript for typo such as line 38 “)”. The manuscript has been reviewed and typos were corrected.

3.  Please provide a citation for “Studies investigating an MS animal model revealed harmful as well as beneficial effects of saturated fats depending on their aliphatic chain length.”  An appropriate citation has been added.

4. Please correct the statement “Serum and red blood cells (RBCs) were immediately centrifuged” as this sentence appears to be incomplete. The sentence has been corrected.

5. The material and method section is short, and most procedures are not explained in detail. Please provide a more detailed description of the performed experiments to increase reproducibility of the results. We have added more details and expanded this section.

6. Did the authors check whether their data show a normal distribution? If so, which test was performed. Due to the relatively large sample sizes compared between MS and healthy participants (N>30), we assumed normality. Therefore, we chose the independent T-test, whereas in the measures comparing sub-groups in which (N<30), we performed the Mann-Whitney non-parametric test.

7. The study design is not without risk. The authors compare nutritional intake between MS patients and healthy controls. However, the clinical disability per se might be an important variable for the study outcome. It would have been elegant to include a group with comparable level of clinical disability suffering from another neurological disorder than MS. However, I am aware that this is a complicated study design. At least the authors should discuss this point. We  have added this important point to the Discussion stating that “It is possible that regardless of MS there is a link between dietary data and physical disability.” We chose not to include in this study an additional neurological condition as we felt it would not necessarily serve as a reliable control group.

8. To what extent is a 24h dietary recall “representative/meaningful”? The dietary information obtained from such questionnaires is vast. It is important, however, to recognize the limitations of such questioning in terms of accuracy. Some participants may not report their exact intake in order to impress the survey administrators or simplify the survey process. For this reason it was important for us to include in this study results from blood tests as well.

9. Please check lines 157-160. We apologize for mistakenly leaving these lines in the manuscript sent, they were deleted.

10. Please provide information regarding the patient’s treatment. Did the treatment affect the outcome of the study? We have added data on patient treatment (Result section 3.1), and now present Supplementary Table 1 showing that there are no significant differences in fatty acid profiles in RBC membranes found between treatment groups. Analysis of blood parameters and dietary intake according to treatment groups did not reveal any significant differences either (data not shown).

11. The authors found lower iron levels in MS patients versus control. However, as stated correctly, Iron is considered an important factor in the pathogenesis of MS, as it may cause neuronal damage by stimulating oxidative stress. This discrepancy should be discussed more in detail. We thank the reviewer for highlighting this point. We have added this in the Discussion.

12. With respect to copper levels, a brief discussion of the cuprizone model would be beneficial. We thank the reviewer for emphasizing this point. We have added this in the Discussion.

13. Please include a recent study (J Neurochem. 2018 Jun;145(6):504-515) addressing the lipid profile in a preclinical MS model. We thank the reviewer for suggesting this study. We have added this in the Discussion.

Round  2

Reviewer 1 Report

The authors by Omer et al, have addressed all my concerns.

Author Response

We would like to thank the reviewer once again for the comments and suggestions which have improved our current manuscript.

Reviewer 2 Report

While the authors have adressed most points, normal data distribution should not be assumed but tested by appropriate procedures. Please revise this part of the manuscript accordingly.

Author Response

We would like to thank the reviewer once again for the comments and suggestions which have improved our current manuscript. We now clearly indicate in the methods section that only age and BMI (presented in Table 1) were assessed by independent T-test. This was done after assessing normality according to the Kolomorov-Smirnov and Shapiro-Wilk tests (e.g. for age p=0.200, p=0.665, accordingly). The additional comparisons between MS and control groups (Tables 2-4), were assessed by Mann-Whitney non-parametric test due to sample sizes of N<30.< p="">

Nutrients EISSN 2072-6643 Published by MDPI AG, Basel, Switzerland RSS E-Mail Table of Contents Alert
Back to Top